# Vision-Language Models are Zero-Shot Reward Models for Reinforcement Learning

**Juan Rocamonde**[†‡]
FAR AI

**Victoriano Montesinos**
Vertebra

**Elvis Nava**
ETH AI Center

**Ethan Perez**[*]
Anthropic

**David Lindner**[*‡]
ETH Zurich

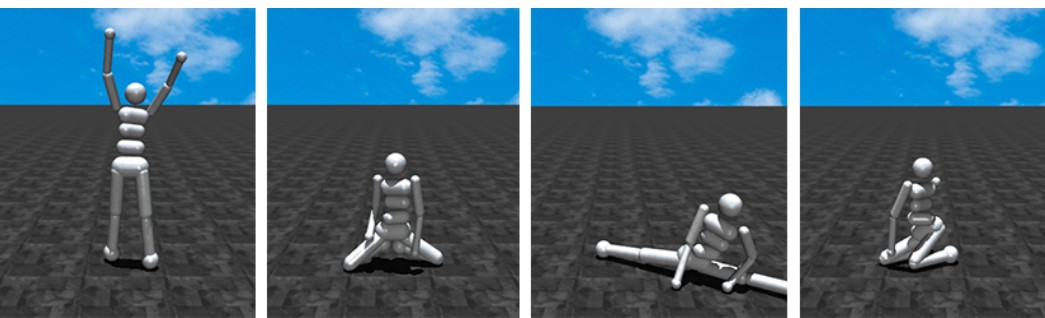

Figure 1: We use CLIP as a reward model to train a MuJoCo humanoid robot to (1) stand with raised arms, (2) sit in a lotus position, (3) do the splits, and (4) kneel on the ground (from left to right). We specify each task using a single sentence text prompt. The prompts are simple (e.g., "a humanoid robot kneeling") and none of these tasks required prompt engineering. See Section 4.3 for details on our experimental setup.

## ABSTRACT

Reinforcement learning (RL) requires either manually specifying a reward function, which is often infeasible, or learning a reward model from a large amount of human feedback, which is often very expensive. We study a more sample-efficient alternative: using pretrained vision-language models (VLMs) as zero-shot reward models (RMs) to specify tasks via natural language. We propose a natural and general approach to using VLMs as reward models, which we call VLM-RMs. We use VLM-RMs based on CLIP to train a MuJoCo humanoid to learn complex tasks without a manually specified reward function, such as kneeling, doing the splits, and sitting in a lotus position. For each of these tasks, we only provide *a single sentence text prompt* describing the desired task with minimal prompt engineering. We provide videos of the trained agents at: https://sites.google.com/view/vlm-rm[1]. We can improve performance by providing a second "baseline" prompt and projecting out parts of the CLIP embedding space irrelevant to distinguish between goal and baseline. Further, we find a strong scaling effect for VLM-RMs: larger VLMs trained with more compute and data are better reward models. The failure modes of VLM-RMs we encountered are all related to known capability limitations of current VLMs, such as limited spatial reasoning ability or visually unrealistic environments that are far off-distribution for the VLM. We find that VLM-RMs are remarkably robust as long as the VLM is large enough. This suggests that future VLMs will become more and more useful reward models for a wide range of RL applications.

---

[†]Additional affiliation: Vertebra

[‡]Correspondence to: juancarlosrocamonde@gmail.com, david.lindner@inf.ethz.ch

[*]Equal contribution

[1]Source code available at https://github.com/AlignmentResearch/vlmrm

# 1 INTRODUCTION

Training reinforcement learning (RL) agents to perform complex tasks in vision-based domains can be difficult, due to high costs associated with reward specification. Manually specifying reward functions for real world tasks is often infeasible, and learning a reward model from human feedback is typically expensive. To make RL more useful in practical applications, it is critical to find a more sample-efficient and natural way to specify reward functions.

One natural approach is to use pretrained vision-language models (VLMs), such as CLIP (Radford et al., 2021) and Flamingo (Alayrac et al., 2022), to provide reward signals based on natural language. However, prior attempts to use VLMs to provide rewards require extensive fine-tuning VLMs (e.g., Du et al., 2023) or complex ad-hoc procedures to extract rewards from VLMs (e.g., Mahmoudieh et al., 2022). In this work, we demonstrate that simple techniques for using VLMs as *zero-shot* language-grounded reward models work well, as long as the chosen underlying model is sufficiently capable. Concretely, we make four key contributions.

First, we **propose VLM-RM**, a general method for using pre-trained VLMs as a reward model for vision-based RL tasks (Section 3). We propose a concrete implementation that uses CLIP as a VLM and cos-similarity between the CLIP embedding of the current environment state and a simple language prompt as a reward function. We can optionally regularize the reward model by providing a "baseline prompt" that describes a neutral state of the environment and partially projecting the representations onto the direction between baseline and target prompts when computing the reward.

Second, we **validate our method in the standard `CartPole` and `MountainCar` RL benchmarks** (Section 4.2). We observe high correlation between VLM-RMs and the ground truth rewards of the environments and successfully train policies to solve the tasks using CLIP as a reward model. Furthermore, we find that the quality of CLIP as a reward model improves if we render the environment using more realistic textures.

Third, we train a **MuJoCo humanoid to learn complex tasks**, including raising its arms, sitting in a lotus position, doing the splits, and kneeling (Figure 1; Section 4.3) using a CLIP reward model derived from single sentence text prompts (e.g., "a humanoid robot kneeling").

Fourth, we **study how VLM-RMs' performance scales** with the size of the VLM, and find that VLM scale is strongly correlated to VLM-RM quality (Section 4.4). In particular, we can only learn the humanoid tasks in Figure 1 with the largest publicly available CLIP model.

Our results indicate that VLMs are powerful zero-shot reward models. While current models, such as CLIP, have important limitations that persist when used as VLM-RMs, we expect such limitations to mostly be overcome as larger and more capable VLMs become available. Overall, VLM-RMs are likely to enable us to train models to perform increasingly sophisticated tasks from human-written task descriptions.

# 2 BACKGROUND

**Partially observable Markov decision processes.** We formulate the problem of training RL agents in vision-based tasks as a partially observable Markov decision process (POMDP). A POMDP is a tuple $(\mathcal{S}, \mathcal{A}, \theta, R, \mathcal{O}, \phi, \gamma, d_0)$ where: $\mathcal{S}$ is the state space; $\mathcal{A}$ is the action space; $\theta(s'|s, a) : \mathcal{S} \times \mathcal{S} \times \mathcal{A} \to [0, 1]$ is the transition function; $R(s, a, s') : \mathcal{S} \times \mathcal{A} \times \mathcal{S} \to \mathbb{R}$ is the reward function; $\mathcal{O}$ is the observation space; $\phi(o|s) : \mathcal{S} \to \Delta(\mathcal{O})$ is the observation distribution; and $d_0(s) : \mathcal{S} \to [0, 1]$ is the initial state distribution.

At each point in time, the environment is in a state $s \in \mathcal{S}$. In each timestep, the agent takes an action $a \in \mathcal{A}$, causing the environment to transition to state $s'$ with probability $\theta(s'|s, a)$. The agent then receives an observation $o$, with probability $\phi(o|s')$ and a reward $r = R(s, a, s')$. A sequence of states and actions is called a trajectory $\tau = (s_0, a_0, s_1, a_1, \dots)$, where $s_i \in \mathcal{S}$, and $a_i \in \mathcal{A}$. The returns of such a trajectory $\tau$ are the discounted sum of rewards $g(\tau; R) = \sum_{t=0} \gamma^t R(s_t, a_t, s_{t+1})$.

The agent's goal is to find a (possibly stochastic) policy $\pi(s|a)$ that maximizes the expected returns $G(\pi) = \mathbb{E}_{\tau(\pi)}[g(\tau(\pi); R)]$. We only consider finite-horizon trajectories, i.e., $|\tau| < \infty$.

**Vision-language models.** We broadly define vision-language models (VLMs; Zhang et al., 2023) as models capable of processing sequences of both language inputs $l \in \mathcal{L}^{\leq n}$ and vision inputs $i \in \mathcal{I}^{\leq m}$. Here, $\mathcal{L}$ is a finite alphabet and $\mathcal{L}^{\leq n}$ contains strings of length less than or equal to $n$, whereas $\mathcal{I}$ is the space of 2D RGB images and $\mathcal{I}^{\leq m}$ contains sequences of images with length less than or equal to $m$.

**CLIP models.** One popular class of VLMs are Contrastive Language-Image Pretraining (CLIP; Radford et al., 2021) encoders. CLIP models consist of a language encoder $\mathrm{CLIP}_L : \mathcal{L}^{\leq n} \to \mathcal{V}$ and an image encoder $\mathrm{CLIP}_I : \mathcal{I} \to \mathcal{V}$ mapping into the same latent space $\mathcal{V} = \mathbb{R}^k$. These encoders are jointly trained via contrastive learning over pairs of images and captions. Commonly CLIP encoders are trained to minimize the cosine distance between embeddings for semantically matching pairs and maximize the cosine distance between semantically non-matching pairs.

## 3 VISION-LANGUAGE MODELS AS REWARD MODELS (VLM-RMs)

This section presents how we can use VLMs as a learning-free (zero-shot) way to specify rewards from natural language descriptions of tasks. Importantly, VLM-RMs avoid manually engineering a reward function or collecting expensive data for learning a reward model.

### 3.1 USING VISION-LANGUAGE MODELS AS REWARDS

Let us consider a POMDP without a reward function $(\mathcal{S}, \mathcal{A}, \theta, \mathcal{O}, \phi, \gamma, d_0)$. We focus on vision-based RL where the observations $o \in \mathcal{O}$ are images. For simplicity, we assume a deterministic observation distribution $\phi(o|s)$ defined by a mapping $\psi(s) : \mathcal{S} \to \mathcal{O}$ from states to image observation. We want the agent to perform a *task* $\mathcal{T}$ based on a natural language description $l \in \mathcal{L}^{\leq n}$. For example, when controlling a humanoid robot (Section 4.3) $\mathcal{T}$ might be the robot kneeling on the ground and $l$ might be the string "a humanoid robot kneeling".

To train the agent using RL, we need to first design a reward function. We propose to use a VLM to provide the reward $R(s)$ as:

$$R_{\mathrm{VLM}}(s) = \mathrm{VLM}(l, \psi(s), c), \tag{1}$$

where $c \in \mathcal{L}^{\leq n}$ is an optional context, e.g., for defining the reward interactively with a VLM. This formulation is general enough to encompass the use of several different kinds of VLMs, including image and video encoders, as reward models.

**CLIP as a reward model.** In our experiments, we chose a CLIP encoder as the VLM. A very basic way to use CLIP to define a reward function is to use cosine similarity between a state's image representation and the natural language task description:

$$R_{\mathrm{CLIP}}(s) = \frac{\mathrm{CLIP}_L(l) \cdot \mathrm{CLIP}_I(\psi(s))}{\|\mathrm{CLIP}_L(l)\| \cdot \|\mathrm{CLIP}_I(\psi(s))\|}. \tag{2}$$

In this case, we do not require a context $c$. We will sometimes call the CLIP image encoder a *state encoder*, as it encodes an image that is a direct function of the POMDP state, and the CLIP language encoder a *task encoder*, as it encodes the language description of the task.

### 3.2 GOAL-BASELINE REGULARIZATION TO IMPROVE CLIP REWARD MODELS

While in the previous section, we introduced a very basic way of using CLIP to define a task-based reward function, this section proposes *Goal-Baseline Regularization* as a way to improve the quality of the reward by projecting out irrelevant information about the observation.

So far, we assumed we only have a task description $l \in \mathcal{L}^{\leq n}$. To apply goal-baseline regularization, we require a second "baseline" description $b \in \mathcal{L}^{\leq n}$. The baseline $b$ is a natural language description of the environment setting in its default state, irrespective of the goal. For example, our baseline description for the humanoid is simply "a humanoid robot," whereas the task description is, e.g., "a humanoid robot kneeling." We obtain the goal-baseline regularized CLIP reward model ($R_{\mathrm{CLIP\text{-}Reg}}$) by projecting our state embedding onto the line spanned by the baseline and task embeddings.

**Definition 1** (Goal-Baseline Regularization). *Given a goal task description $l$ and baseline description $b$, let $\mathbf{g} = \frac{CLIP_L(l)}{\|CLIP_L(l)\|}$, $\mathbf{b} = \frac{CLIP_L(b)}{\|CLIP_L(b)\|}$, $\mathbf{s} = \frac{CLIP_I(\psi(s))}{\|CLIP_I(\psi(s))\|}$ be the normalized encodings, and $L$ be the line spanned by $\mathbf{b}$ and $\mathbf{g}$. The goal-baseline regularized reward function is given by*

$$R_{CLIP\text{-}Reg}(s) = 1 - \frac{1}{2}\|\alpha \operatorname{proj}_L \mathbf{s} + (1-\alpha)\mathbf{s} - \mathbf{g}\|_2^2, \tag{3}$$

*where $\alpha$ is a parameter to control the regularization strength.*

In particular, for $\alpha = 0$, we recover our initial CLIP reward function $R_{\text{CLIP}}$. On the other hand, for $\alpha = 1$, the projection removes all components of $\mathbf{s}$ orthogonal to $\mathbf{g} - \mathbf{b}$.

Intuitively, the direction from $\mathbf{b}$ to $\mathbf{g}$ captures the change from the environment's baseline to the target state. By projecting the reward onto this direction, we directionally remove irrelevant parts of the CLIP representation. However, we can not be sure that the direction really captures all relevant information. Therefore, instead of using $\alpha = 1$, we treat it as a hyperparameter. However, we find the method to be relatively robust to changes in $\alpha$ with most intermediate values being better than 0 or 1.

### 3.3 RL WITH CLIP REWARD MODEL

We can now use VLM-RMs as a drop-in replacement for the reward signal in RL. In our implementation, we use the Deep Q-Network (DQN; Mnih et al., 2015) or Soft Actor-Critic (SAC; Haarnoja et al., 2018) RL algorithms. Whenever we interact with the environment, we store the observations in a replay buffer. In regular intervals, we pass a batch of observations from the replay buffer through a CLIP encoder to obtain the corresponding state embeddings. We can then compute the reward function as cosine similarity between the state embeddings and the task embedding which we only need to compute once. Once we have computed the reward for a batch of interactions, we can use them to perform the standard RL algorithm updates. Appendix C contains more implementation details and pseudocode for our full algorithm in the case of SAC.

## 4 EXPERIMENTS

We conduct a variety of experiments to evaluate CLIP as a reward model with and without goal-baseline regularization. We start with simple control tasks that are popular RL benchmarks: `CartPole` and `MountainCar` (Section 4.2). These environments have a ground truth reward function and a simple, well-structured state space. We find that our reward models are highly correlated with the ground truth reward function, with this correlation being greatest when applying goal-baseline regularization. Furthermore, we find that the reward model's outputs can be significantly improved by making a simple modification to make the environment's observation function more realistic, e.g., by rendering the mountain car over a mountain texture.

We then move on to our main experiment: controlling a simulated humanoid robot (Section 4.3). We use CLIP reward models to specify tasks from short language prompts; several of these tasks are challenging to specify manually. We find that these zero-shot CLIP reward models are sufficient for RL algorithms to learn most tasks we attempted with little to no prompt engineering or hyperparameter tuning.

Finally, we study the scaling properties of the reward models by using CLIP models of different sizes as reward models in the humanoid environment (Section 4.4). We find that larger CLIP models are significantly better reward models. In particular, we can only successfully learn the tasks presented in Figure 1 when using the largest publicly available CLIP model.

**Experiment setup.** We extend the implementation of the DQN and SAC algorithm from the `stable-baselines3` library (Raffin et al., 2021) to compute rewards from CLIP reward models instead of from the environment. As shown in Algorithm 1 for SAC, we alternate between environment steps, computing the CLIP reward, and RL algorithm updates. We run the RL algorithm updates on a single NVIDIA RTX A6000 GPU. The environment simulation runs on CPU, but we perform rendering and CLIP inference distributed over 4 NVIDIA RTX A6000 GPUs.

We provide the code to reproduce our experiments in the supplementary material. We discuss hyperparameter choices in Appendix C, but we mostly use standard parameters from

`stable-baselines3`. Appendix C also contains a table with a full list of prompts for our experiments, including both goal and baseline prompts when using goal-baseline regularization.

## 4.1 HOW CAN WE EVALUATE VLM-RMS?

Evaluating reward models can be difficult, particularly for tasks for which we do not have a ground truth reward function. In our experiments, we use 3 types of evaluation: (i) evaluating policies using ground truth reward; (ii) comparing reward functions using EPIC distance; (iii) human evaluation.

**Evaluating policies using ground truth reward.** If we have a ground truth reward function for a task such as for the `CarPole` and `MountainCar`, we can use it to evaluate policies. For example, we can train a policy using a VLM-RM and evaluate it using the ground truth reward. This is the most popular way to evaluate reward models in the literature and we use it for environments where we have a ground-truth reward available.

**Comparing reward functions using EPIC distance.** The "Equivalent Policy-Invariant Comparison" (EPIC; Gleave et al., 2021) distance compares two reward functions without requiring the expensive policy training step. EPIC distance is provably invariant on the equivalence class of reward functions that induce the same optimal policy. We consider only goal-based tasks, for which the EPIC is distance particularly easy to compute. In particular, a low EPIC distance between the CLIP reward model and the ground truth reward implies that the CLIP reward model successfully separates goal states from non-goal states. Appendix A discusses in more detail how we compute the EPIC distance in our case, and how we can intuitively interpret it for goal-based tasks.

**Human evaluation.** For tasks without a ground truth reward function, such as all humanoid tasks in Figure 1, we need to perform human evaluations to decide whether our agent is successful. We define "success rate" as the percentage of trajectories in which the agent successfully performs the task in at least $50\%$ of the timesteps. For each trajectory, we have a single rater[2] label how many timesteps were spent successfully performing the goal task, and use this to compute the success rate. However, human evaluations can also be expensive, particularly if we want to evaluate many different policies, e.g., to perform ablations. For such cases, we additionally collect a dataset of human-labelled states for each task, including goal states and non-goal states. We can then compute the EPIC distance with these binary human labels. Empirically, we find this to be a useful proxy for the reward model quality which correlates well with the performance of a policy trained using the reward model.

For more details on our human evaluation protocol, we refer to Appendix B. Our human evaluation protocol is very basic and might be biased. Therefore, we additionally provide videos of our trained agents at `https://sites.google.com/view/vlm-rm`.

## 4.2 CAN VLM-RMS SOLVE CLASSIC CONTROL BENCHMARKS?

As an initial validation of our methods, we consider two classic control environments: `CartPole` and `MountainCar`, implemented in OpenAI Gym (Brockman et al., 2016). In addition to the default `MountainCar` environment, we also consider a version with a modified rendering method that adds textures to the mountain and the car so that it resembles the setting of "a car at the peak of a mountain" more closely (see Figure 2). This environment allows us to test whether VLM-RMs work better in visually "more realistic" environments.

To understand the rewards our CLIP reward models provide, we first analyse plots of their reward landscape. In order to obtain a simple and interpretable visualization figure, we plot CLIP rewards against a one-dimensional state space parameter, that is directly related to the completion of the task. For the `CartPole` (Figure 2a) we plot CLIP rewards against the angle of the pole, where the ideal position is at angle $0$. For the (untextured and textured) `MountainCar` environments Figures 2b and 2c, we plot CLIP rewards against the position of the car along the horizontal axis, with the goal location being around $x = 0.5$.

---

[2]One of the author.

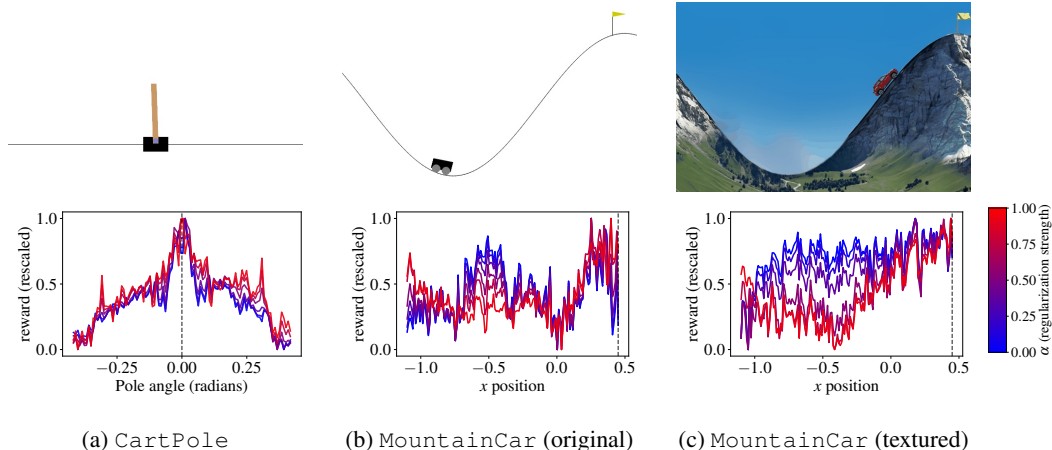

(a) `CartPole`          (b) `MountainCar` (original)          (c) `MountainCar` (textured)

Figure 2: We study the CLIP reward landscape in two classic control environments: `CartPole` and `MountainCar`. We plot the CLIP reward as a function of the pole angle for the `CartPole` (a) and as a function of the x position for the `MountainCar` (b,c). We mark the respective goal states with a vertical line. The line color encodes different regularization strengths $\alpha$. For the `CartPole`, the maximum reward is always when balancing the pole and the regularization has little effect. For the `MountainCar`, the agent obtains the maximum reward on top of the mountain. But, the reward landscape is much more well-behaved when the environment has textures and we add goal-baseline regularization – this is consistent with our results when training policies.

Figure 2a shows that CLIP rewards are well-shaped around the goal state for the `CartPole` environment, whereas Figure 2b shows that CLIP rewards for the default `MountainCar` environment are poorly shaped, and might be difficult to learn from, despite still having roughly the right maximum.

We conjecture that zero-shot VLM-based rewards work better in environments that are more "photorealistic" because they are closer to the training distribution of the underlying VLM. Figure 2c shows that if, as described earlier, we apply custom textures to the `MountainCar` environment, the CLIP rewards become well-shaped when used in concert with the goal-baseline regularization technique. For larger regularization strength $\alpha$, the reward shape resembles the slope of the hill from the environment itself – an encouraging result.

We then train agents using the CLIP rewards and goal-baseline regularization in all three environments, and achieve 100% task success rate in both environments (`CartPole` and textured `MountainCar`) for most $\alpha$ regularization strengths. Without the custom textures, we are not able to successfully train an agent on the mountain car task, which supports our hypothesis that the environment visualization is too abstract.

The results show that both and regularized CLIP rewards are effective in the toy RL task domain, with the important caveat that CLIP rewards are only meaningful and well-shaped for environments that are photorealistic enough for the CLIP visual encoder to interpret correctly.

### 4.3 CAN VLM-RMS LEARN COMPLEX, NOVEL TASKS IN A HUMANOID ROBOT?

Our primary goal in using VLM-RMs is to learn tasks for which it is difficult to specify a reward function manually. To study such tasks, we consider the `Humanoid-v4` environment implemented in the MuJoCo simulator (Todorov et al., 2012).

The standard task in this environment is for the humanoid robot to stand up. For this task, the environment provides a reward function based on the vertical position of the robot's center of mass. We consider a range of additional tasks for which no ground truth reward function is available, including kneeling, sitting in a lotus position, and doing the splits. For a full list of tasks we tested, see Table 1. Appendix C presents more detailed task descriptions and the full prompts we used.

| Task | Success Rate |
|------|--------------|
| Kneeling | **100%** |
| Lotus position | **100%** |
| Standing up | **100%** |
| Arms raised | **100%** |
| Doing splits | **100%** |
| Hands on hips | 64% |
| Standing on one leg | 0% |
| Arms crossed | 0% |

Table 1: We successfully learned 5 out of 8 tasks we tried for the humanoid robot (cf. Figure 1). For each task, we evaluate the checkpoint with the highest CLIP reward over 4 random seeds. We show a human evaluator 100 trajectories from the agent and ask them to label how many timesteps were spent successfully performing the goal task. Then, we label an episode as a success if the agent is in the goal state at least 50% of the timesteps. The success rate is the fraction of trajectories labelled as successful. We provide more details on the evaluation as well as more fine-grained human labels in Appendix B and videos of the agents' performance at https://sites.google.com/view/vlm-rm.

| Camera Angle | Textures | Success Rate |
|--------------|----------|--------------|
| Original | Original | 36% |
| Original | Modified | 91% |
| Modified | Modified | **100%** |

(a) Original     (b) Modified textures     (c) Modified textures & camera angle

Figure 3: We test the effect of our modifications to the standard Humanoid-v4 environment on the kneeling task. We compare the original environment (a) to modifying the textures (b) and the camera angle (c). We find that modifying the textures to be more realistic is crucial to making the CLIP reward model work. Moving the camera to give a better view of the humanoid helps too, but is less critical in this task.

We make two modifications to the default `Humanoid-v4` environment to make it better suited for our experiments. (1) We change the colors of the humanoid texture and the environment background to be more realistic (based on our results in Section 4.2 that suggest this should improve the CLIP encoder). (2) We move the camera to a fixed position pointing at the agent slightly angled down because the original camera position that moves with the agent can make some of our tasks impossible to evaluate. We ablate these changes in Figure 3, finding the texture change is critical and repositioning the camera provides a modest improvement.

Table 1 shows the human-evaluated success rate for all tasks we tested. We solve 5 out of 8 tasks we tried with minimal prompt engineering and tuning. For the remaining 3 tasks, we did not get major performance improvements with additional prompt engineering and hyperparameter tuning, and we hypothesize these failures are related to capability limitations in the CLIP model we use. We invite the reader to evaluate the performance of the trained agents themselves by viewing videos at https://sites.google.com/view/vlm-rm.

The three tasks that the agent does not obtain perfect performance for are "hands on hips", "standing on one leg", and "arms crossed". We hypothesize that "standing on one leg" is very hard to learn or might even be impossible in the MuJoCo physics simulation because the humanoid's feet are round. The goal state for "hands on hips" and "arms crossed" is visually similar to a humanoid standing and we conjecture the current generation of CLIP models are unable to discriminate between such subtle differences in body pose.

While the experiments in Table 1 use no goal-baseline regularization (i.e., $\alpha = 0$), we separately evaluate goal-baseline regularization for the kneeling task. Figure 4a shows that $\alpha \neq 0$ improves the reward model's EPIC distance to human labels, suggesting that it would also improve performance on the final task, we might need a more fine-grained evaluation criterion to see that.

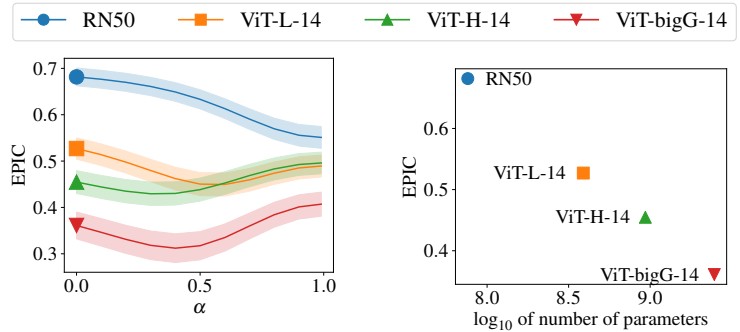

(a) Goal-baseline regularization for different model sizes.

(b) Reward model performance by VLM training compute ($\alpha = 0$).

(c) Human-evaluated success rate (over 2 seeds).

Figure 4: VLMs become better reward models with VLM model scale. We evaluate the humanoid kneeling task for different VLM model sizes. We evaluate the EPIC distance between the CLIP rewards and human labels (a and b) and the human-evaluated success rate of an agent trained using differently sized CLIP reward models (c). We see a strong positive effect of model scale on VLM-RM quality. In particular, (c) shows we are only able to learn the kneeling task using the largest CLIP model publically available, whereas (b) shows there is a smooth improvement in EPIC distance compared to human labels. (a) shows that goal-baseline regularization improves the reward model across model sizes but it is more impactful for small models.

## 4.4 How do VLM-RMs Scale with VLM Model Size?

Finally, we investigate the effect of the scale of the pre-trained VLM on its quality as a reward model. We focus on the "kneeling" task and consider 4 different large CLIP models: the original CLIP `RN50` (Radford et al., 2021), and the `ViT-L-14`, `ViT-H-14`, and `ViT-bigG-14` from OpenCLIP (Cherti et al., 2023) trained on the LAION-5B dataset (Schuhmann et al., 2022).

In Figure 4a we evaluate the EPIC distance to human labels of CLIP reward models for the four model scales and different values of $\alpha$, and we evaluate the success rate of agents trained using the four models. The results clearly show that VLM model scale is a key factor in obtaining good reward models. We detect a clear positive trend between model scale, and the EPIC distance of the reward model from human labels. On the models we evaluate, we find the EPIC distance to human labels is close to log-linear in the size of the CLIP model (Figure 4b).

This improvement in EPIC distance translates into an improvement in success rate. In particular, we observe a sharp phase transition between the `ViT-H-14` and `VIT-bigG-14` CLIP models: we can only learn the kneeling task successfully when using the `VIT-bigG-14` model and obtain 0% success rate for all smaller models (Figure 4c). Notably, the reward model improves smoothly and predictably with model scale as measured by EPIC distance. However, predicting the exact point where the RL agent can successfully learn the task is difficult. This is a common pattern in evaluating large foundation models, as observed by Ganguli et al. (2022).

## 5 Related Work

Foundation models (Bommasani et al., 2021) trained on large scale data can learn remarkably general and transferable representations of images, language, and other kinds of data, which makes them useful for a large variety of downstream tasks. For example, pre-trained vision-language encoders, such as CLIP (Radford et al., 2021), have been used far beyond their original scope, e.g., for image generation (Ramesh et al., 2022; Patashnik et al., 2021; Nichol et al., 2021), robot control (Shridhar et al., 2022; Khandelwal et al., 2022), or story evaluation (Matiana et al., 2021).

Reinforcement learning from human feedback (RLHF; Christiano et al., 2017) is a critical step in making foundation models more useful (Ouyang et al., 2022). However, collecting human feedback is expensive. Therefore, using pre-trained foundation models themselves to obtain reward signals for RL finetuning has recently emerged as a key paradigm in work on large language models (Bai

et al., 2022). Some approaches only require a small amount of natural language feedback instead of a whole dataset of human preferences (Scheurer et al., 2022; 2023; Chen et al., 2023). However, similar techniques have yet to be adopted by the broader RL community.

While some work uses language models to compute a reward function from a structured environment representation (Xie et al., 2023; Ma et al., 2023), many RL tasks are visual and require using VLMs instead. Sumers et al. (2023) use generative VLMs to relabel the goal of agent trajectories for hindsight experience replay, but not for specifying rewards. Cui et al. (2022) use CLIP to provide rewards for robotic manipulation tasks given a goal image. However, they only show limited success when using natural language descriptions to define goals, which is the focus of our work. Mahmoudieh et al. (2022) are the first to successfully use CLIP encoders as a reward model conditioned on language task descriptions in robotic manipulation tasks. However, to achieve this, the authors need to explicitly fine-tune the CLIP image encoder on a carefully crafted dataset for a robotics task. Instead, we focus on leveraging CLIP's zero-shot ability to specify reward functions, which is significantly more sample-efficient and practical. Fan et al. (2022) train a CLIP model to provide a reward signal in Minecraft environments. But, that approach requires a lot of labeled, environment-specific data. Du et al. (2023) finetune a Flamingo VLM (Alayrac et al., 2022) to act as a "success detector" for vision-based RL tasks tasks. However, they do not train RL policies using these success detectors, leaving open the question of how robust they are under optimization pressure. Concurrently to our work, Sontakke et al. (2023) successfully use a VLM to provide reward signals for RL agents in robotics settings. However, they focus on specifying the reward with video demonstrations and only show basic results with natural language task descriptions.

In contrast to these works, we do not require any finetuning to use CLIP as a reward model, and we successfully train RL policies to achieve a range of complex tasks that do not have an easily-specified ground truth reward function.

## 6  CONCLUSION

We introduced a method to use vision-language models (VLMs) as reward models for reinforcement learning (RL), and implemented it using CLIP as a reward model and standard RL algorithms. We used VLM-RMs to solve classic RL benchmarks and to learn to perform complicated tasks using a simulated humanoid robot. We observed a strong scaling trend with model size, which suggests that future VLMs are likely to be useful as reward models in an even broader range of tasks.

**Limitations.** Fundamentally, our approach relies on the reward model generalizing from a text description to a reward function that captures what a human intends the agent to do. Although the concrete failure cases we observed are likely specific to the CLIP models we used and may be solved by more capable models, some problems will persist. The resulting reward model will be misspecified if the text description does not contain enough information about what the human intends or the VLM generalizes poorly. While we expect future VLMs to generalize better, the risk of the reward model being misspecified grows for more complex tasks, that are difficult to specify in a single language prompt. Therefore, when using VLM-RMs in practice it will be crucial to use independent monitoring to ensure agents trained from automated feedback act as intended. For complex tasks, it will be prudent to use a multi-step reward specification, e.g., by using a VLM capable of having a dialogue with the user about specifying the task.

**Future Work.** There are many possible extensions of our approach that may improve performance but were not necessary in our tasks. For example, finetuning VLMs for specific environments is a natural next step to make them more useful as reward models. To move beyond goal-based supervision, future VLM-RMs could encode videos instead of images. To move towards specifying more complex tasks, future VLM-RMs could use dialogue-enabled VLMs.

For practical applications, it will be important to ensure robustness and safety of the reward model. Our work can serve as a basis for studying the safety implications of VLM-RMs. For instance, future work could investigate the robustness of VLM-RMs against optimization pressure by RL agents.

More broadly, we believe VLM-RMs open up exciting avenues for future research to build useful agents on top of pre-trained models, such as building language model agents and real world robotic controllers for tasks where we do not have a reward function available.

AUTHOR CONTRIBUTIONS

**Juan Rocamonde** designed and implemented the experimental infrastructure, ran most experiments, analyzed results, and wrote large parts of the paper.

**Victoriano Montesinos** implemented parallelized rendering and training to enable using larger CLIP models, implemented and ran many experiments, and performed the human evaluations.

**Elvis Nava** advised on experiment design, implemented and ran some of the experiments, and wrote large parts of the paper.

**Ethan Perez** proposed the original project and advised on research direction and experiment design.

**David Lindner** implemented and ran early experiments with the humanoid robot, wrote large parts of the paper, and led the project.

ACKNOWLEDGMENTS

We thank Adam Gleave for valuable discussions throughout the project and detailed feedback on early drafts, Jérémy Scheurer and Nora Belrose for helpful feedback early on, Adrià Garriga-Alonso for help with running experiments, and Xander Balwit for help with editing the paper.

We are grateful for funding received by Open Philanthropy, Manifund, the ETH AI Center, Swiss National Science Foundation (B.F.G. CRSII5-173721 and 315230 189251), ETH project funding (B.F.G. ETH-20 19-01), and the Human Frontiers Science Program (RGY0072/2019).

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

## A    COMPUTING AND INTERPRETING EPIC DISTANCE

Our experiments all have goal-based ground truth reward functions, i.e., they give high reward if a goal state is reached and low reward if not. This section discusses how this helps to estimate EPIC distance between reward functions more easily. As a side-effect, this gives us an intuitive understanding of EPIC distance in our context. First, let us define EPIC distance.

**Definition 2** (EPIC distance; Gleave et al. (2021)). *The Equivalent-Policy Invariant Comparison (EPIC) distance between reward functions $R_1$ and $R_2$ is:*

$$D_{EPIC} = \frac{1}{\sqrt{2}}\sqrt{1 - \rho(\mathcal{C}(R_1), \mathcal{C}(R_2))} \tag{4}$$

*where $\rho(\cdot, \cdot)$ is the Pearson correlation w.r.t a given distribution over transitions, and $\mathcal{C}(R)$ is the canonically shaped reward, defined as:*

$$\mathcal{C}(R)(s, a, s') = R(s, a, s') + \mathbb{E}[\gamma R(s', A, S') - R(s, a, S') - \gamma R(S, A, S')].$$

For goal-based tasks, we have a reward function $R(s, a, s') = R(s') = \mathbb{1}_{S_{\mathcal{T}}}(s')$, which assigns a reward of 1 to "goal" states and 0 to "non-goal" states based on the task $\mathcal{T}$. In our experiments, we focus on goal-based tasks because they are most straightforward to specify using image-text encoder VLMs. We expect future models to be able to provide rewards for a more general class of tasks, e.g., using video encoders. For goal-based tasks computing the EPIC distance is particularly convenient.

**Lemma 1** (EPIC distance for CLIP reward model). *Let $(CLIP_I, CLIP_L)$ be a pair of state and task encoders as defined in Section 3.1. Let $R_{CLIP}$ be the CLIP reward function as defined in eq. (2), and $R(s) = \mathbb{1}_{S_{\mathcal{T}}}(s)$ be the ground truth reward function, where $S_{\mathcal{T}}$ is the set of goal states for our task $l$. Let $\mu$ be a probability measure in the state space, let $\rho(\cdot, \cdot)$ be the Pearson correlation under measure $\mu$ and $\mathrm{Var}(\cdot)$ the variance under measure $\mu$. Then, we can compute the EPIC distance of a CLIP reward model and the ground truth reward as:*

$$D_{\mathrm{EPIC}} = \frac{1}{\sqrt{2}}\sqrt{1 - \rho(R_{CLIP}, R)},$$

$$\rho(R_{CLIP}, R) = \frac{\sqrt{\mathrm{Var}(R)}}{\sqrt{\mathrm{Var}(R_{CLIP})}}\left(CLIP_L(l) \cdot \left(\int_{S_{\mathcal{T}}} CLIP_I(\psi(s))\mathrm{d}\mu(s) - \int_{S_{\mathcal{T}}^C} CLIP_I(\psi(s))\mathrm{d}\mu(s)\right)\right),$$

*where $S_{\mathcal{T}}^C = \mathcal{S} \setminus S_{\mathcal{T}}$.*

*Proof.* First, note that for reward functions where the reward of a transition $(s, a, s')$ only depends on $s'$, the canonically-shaped reward simplifies to:

$$\mathcal{C}(R)(s') = R(s') + \gamma\mathbb{E}[R(S')] - \mathbb{E}[R(S')] - \gamma\mathbb{E}[R(S')]$$
$$= R(s') - \mathbb{E}[R(S')].$$

Hence, because the Pearson correlation is location-invariant, we have

$$\rho(\mathcal{C}(R_1), \mathcal{C}(R_2)) = \rho(R_1, R_2).$$

Let $p = \mathbb{P}(Y = 1)$ and recall that $\mathrm{Var}[Y] = p(1-p)$. Then, we can simplify the Pearson correlation between continuous variable $X$ and Bernoulli random variable $Y$ as:

$$\begin{aligned}\rho(X, Y) &:= \frac{\mathrm{Cov}[X, Y]}{\sqrt{\mathrm{Var}[X]}\sqrt{\mathrm{Var}[Y]}} = \frac{\mathbb{E}[XY] - \mathbb{E}[X]\mathbb{E}[Y]}{\sqrt{\mathrm{Var}[X]}\sqrt{\mathrm{Var}[Y]}} = \frac{\mathbb{E}[X|Y=1]p - \mathbb{E}[X]p}{\sqrt{\mathrm{Var}[X]}\sqrt{\mathrm{Var}[Y]}}\\ &= \frac{\mathbb{E}[X|Y=1]p - \mathbb{E}[X|Y=1]p^2 - \mathbb{E}[X|Y=0](1-p)p}{\sqrt{\mathrm{Var}[X]}\sqrt{\mathrm{Var}[Y]}}\\ &= \frac{\mathbb{E}[X|Y=1]p(1-p) - \mathbb{E}[X|Y=0](1-p)p}{\sqrt{\mathrm{Var}[X]}\sqrt{\mathrm{Var}[Y]}}\\ &= \frac{\sqrt{\mathrm{Var}[Y]}}{\sqrt{\mathrm{Var}[X]}}\left(\mathbb{E}[X|Y=1] - \mathbb{E}[X|Y=0]\right).\end{aligned}$$

Combining both results, we obtain that:

$$\rho(\mathcal{C}(R_{\mathrm{CLIP}}), \mathcal{C}(R)) = \frac{\sqrt{\mathrm{Var}(R)}}{\sqrt{\mathrm{Var}(R_{\mathrm{CLIP}})}} \left( \mathbb{E}_{\mathcal{S}_{\mathcal{T}}}[R_{\mathrm{CLIP}}] - \mathbb{E}_{\mathcal{S}_{\mathcal{T}}^C}[R_{\mathrm{CLIP}}] \right)$$

$\square$

If our ground truth reward function is of the form $R(s) = \mathbb{1}_{S_{\mathcal{T}}}(s)$ and we denote $\pi_R^*$ as the optimal policy for reward function $R$, then the quality of $\pi_{R_{\mathrm{CLIP}}}^*$ depends entirely on the Pearson correlation $\rho(R_{\mathrm{CLIP}}, R)$. If $\rho(R_{\mathrm{CLIP}}, R)$ is positive, the cosine similarity of the task embedding with embeddings for goal states $s \in S_{\mathcal{T}}$ is higher than that with embeddings for non-goal states $s \in S_{\mathcal{T}}^C$. Intuitively, $\rho(R_{\mathrm{CLIP}}, R)$ is a measure of how well CLIP separates goal states from non-goal states.

In practice, we use Lemma 1 to evaluate EPIC distance between a CLIP reward model and a ground truth reward function.

Note that the EPIC distance depends on a state distribution $\mu$ (see Gleave et al. (2021) for further discussion). In our experiment, we use either a uniform distribution over states (for the toy RL environments) or the state distribution induced by a pre-trained expert policy (for the humanoid experiments). More details on how we collected the dataset for evaluating EPIC distances can be found in the Appendix B.

## B    HUMAN EVALUATION

Evaluation on tasks for which we do not have a reward function was done manually by one of the authors, depending on the amount of time the agent met the criteria listed in Table 2. See Figures 5 and 6 for the raw labels obtained about the agent performance.

We further evaluated the impact of goal-baseline regularization on the humanoid tasks that did not succeed in our experiments with $\alpha = 0$, cf. Figure 8. In these cases, goal baseline regularization does not improve performance. Together with the results in Figure 4a, this could suggest that goal-baseline regularization is more useful for smaller CLIP models than for larger CLIP models. Alternatively, it is possible that the improvements to the reward model obtained by goal-baseline regularization are too small to lead to noticeable performance increases in the trained agents for the failing humanoid tasks. Unfortunately, a more thorough study of this was infeasible due to the cost associated with human evaluations.

Our second type of human evaluation is to compute the EPIC distance of a reward model to a pre-labelled set of states. To create a dataset for these evaluations, we select all checkpoints from the training run with the highest VLM-RM reward of the largest and most capable VLM we used. We then collect rollouts from each checkpoint and collect the images across all timesteps and rollouts into a single dataset. We then have a human labeller (again an author of this paper) label each image according to whether it represents the goal state or not, using the same criteria from Table 2. We use such a dataset for Figure 4. Figure 7 shows a more detailed breakdown of the EPIC distance for different model scales.

## C    IMPLEMENTATION DETAILS & HYPERPARAMETER CHOICES

In this section, we describe implementation details for both our toy RL environment experiments and the humanoid experiments, going into further detail on the experiment design, any modifications we make to the simulated environments, and the hyperparameters we choose for the RL algorithms we use.

Algorithm 1 shows pseudocode of how we integrate computing CLIP rewards with a batched RL algorithm, in this case SAC.

### C.1    CLASSIC CONTROL ENVIRONMENTS

**Environments.**    We use the standard `CartPole` and `MountainCar` environments implemented in Gym, but remove the termination conditions. Instead the agent receives a negative reward

| Task | Condition |
|------|-----------|
| Kneeling | Agent must be kneeling with both knees touching the floor. Agent must not be losing balance nor kneeling in the air. |
| Lotus position | Agent seated down in the lotus position. Both knees are on the floor and facing outwards, while feet must be facing inwards. |
| Standing up | Agent standing up without falling. |
| Arms raised | Agent standing up with both arms raised. |
| Doing splits | Agent on the floor doing the side splits. Legs are stretched on the floor. |
| Hands on hips | Agent standing up with both hands on the base of the hips. Hands must not be on the chest. |
| Arms crossed | Agent standing up with its arms crossed. If the agent has its hands just touching but not crossing, it is not considered valid. |
| Standing on one leg | Agent standing up touching the floor only with one leg and without losing balance. Agent must not be touching the floor with both feet. |

Table 2: Criteria used to evaluate videos of rollouts generated by the policies trained using CLIP rewards on the humanoid environment. A rollout is considered a success if the agent satisfies the condition for the task at least 50% of the timesteps, and a failure otherwise.

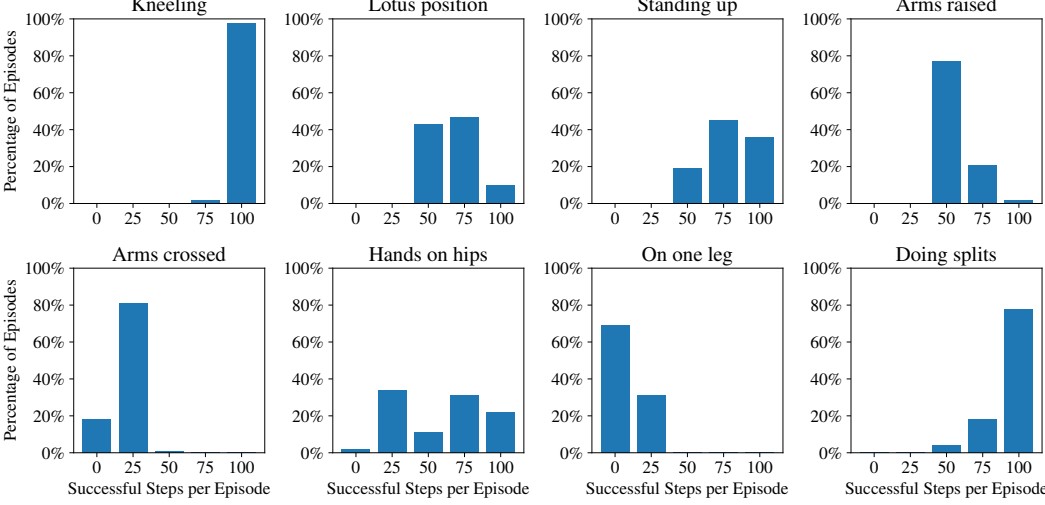

Figure 5: Raw results of our human evaluations. Each histogram is over 100 trajectories sampled from the final policy. One human rater labeled each trajectory in one of five buckets according to whether the agent performs the task correctly $0, 25, 50, 75,$ or $100$ steps out of an episode length of $100$. To compute the success rate in the main paper, we consider all values above $50$ steps as a "success".

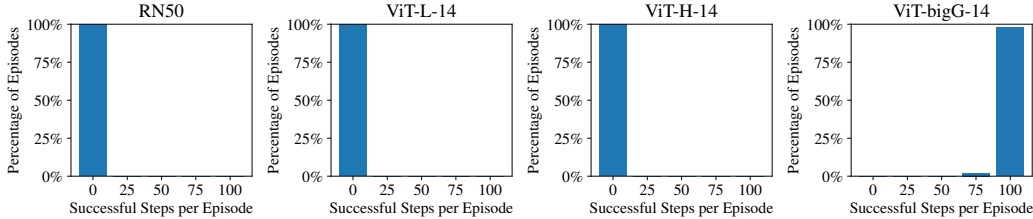

Figure 6: Raw results of our human evaluations for the model scaling experiments. The histograms are computed the same way as in Figure 5, but the agents were trained for differently sized CLIP models on the humanoid "kneeling" task. As the aggregated results in Figure 4c in the main paper suggest, there is a stark difference between the agent trained using the `ViT-H-14` model and the `ViT-bigG-14` model.

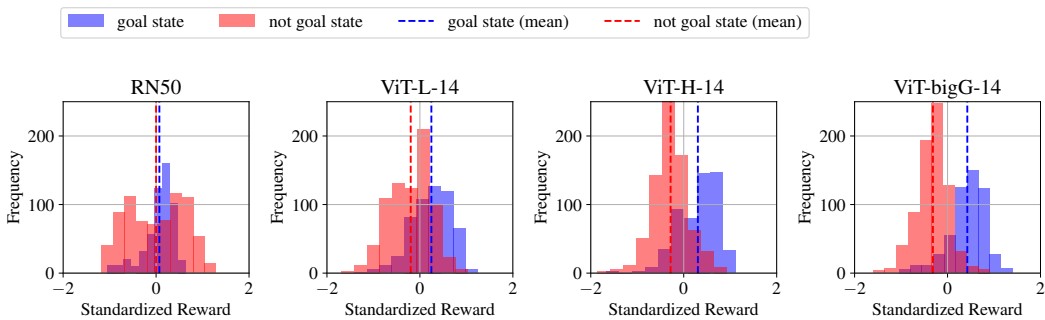

Figure 7: The rewards distributions of (human labelled) goal states vs. non-goal states become more separated with the scale of the VLM. We show histograms of the CLIP rewards for differently labelled states in the humanoid "kneeling" task. The separation between the dotted lines, showing the average of each distribution, is the Pearson correlation described in Appendix A. This provides a clear visual representation of the VLM's capability.

| | Task | Goal Prompt | Baseline Prompt |
|---|---|---|---|
| | `CartPole` | "pole vertically upright on top of the cart" | "pole and cart" |
| | `MountainCar` | "a car at the peak of the mountain, next to the yellow flag" | "a car in the mountain" |
| `Humanoid` | Kneeling | "a humanoid robot kneeling" | "a humanoid robot" |
| | Lotus position | "a humanoid robot seated down, meditating in the lotus position" | "a humanoid robot" |
| | Standing up | "a humanoid robot standing up" | "a humanoid robot" |
| | Arms raised | "a humanoid robot standing up, with both arms raised" | "a humanoid robot" |
| | Doing splits | "a humanoid robot practicing gymnastics, doing the side splits" | "a humanoid robot" |
| | Hands on hips | "a humanoid robot standing up with hands on hips" | "a humanoid robot" |
| | Arms crossed | "a humanoid robot standing up, with its arms crossed" | "a humanoid robot" |
| | Standing on one leg | "a humanoid robot standing up on one leg" | "a humanoid robot" |

Table 3: Goal and baseline prompts for each environment and task. Note that we did not perform prompt engineering, these are the first prompts we tried for every task.

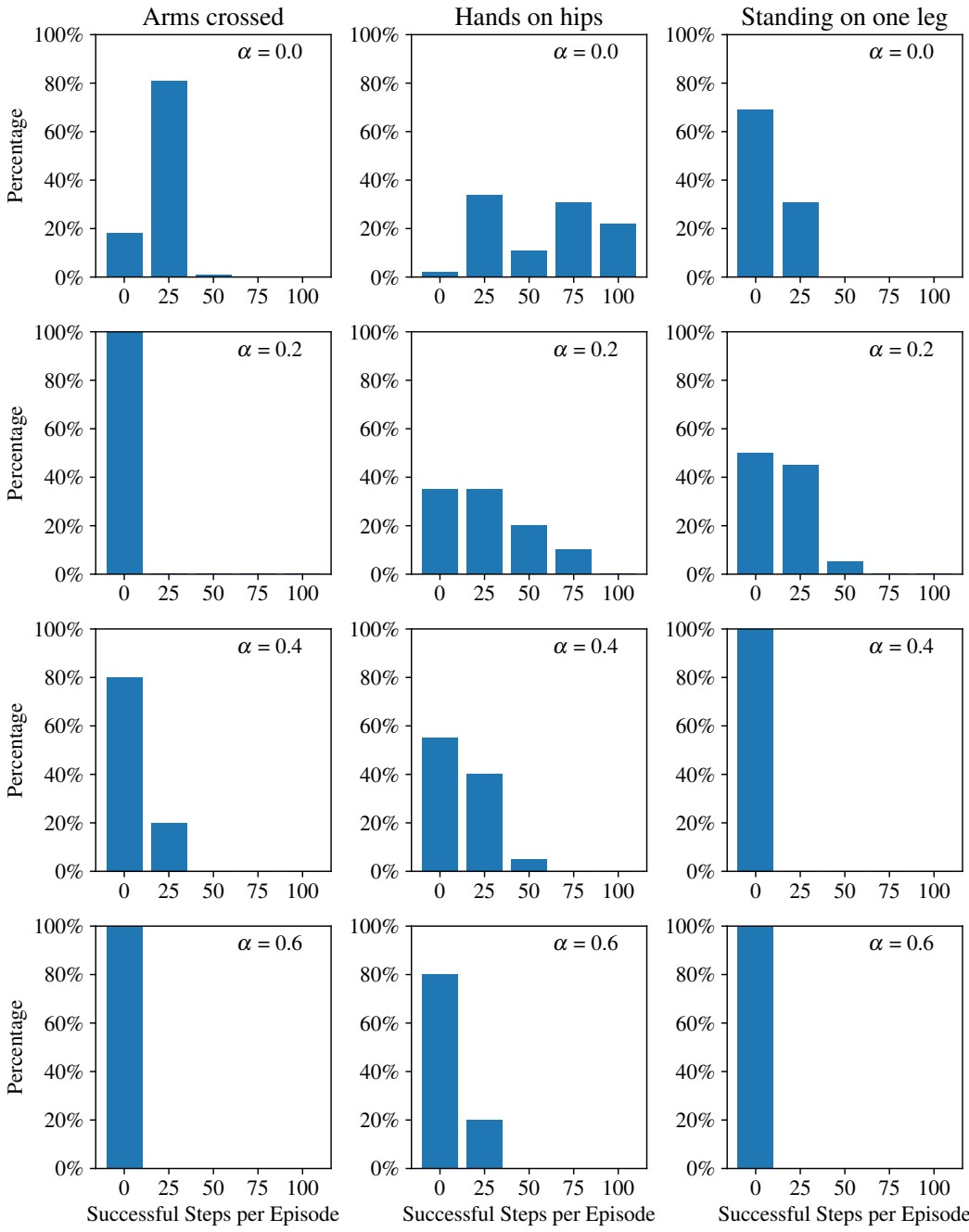

Figure 8: Human evaluations for evaluating goal-baseline regularization in humanoid tasks. The histograms are computed the same way as in Figure 5. We show the humanoid "arms crossed", "hands on hips", and "standing on one leg" tasks that failed in our experiments with $\alpha = 0$. Each column shows one of the tasks and the rows show regularization strength values $\alpha = 0.0, 0.2, 0.4, 0.6$. The performance for $\alpha = 0$ and $\alpha = 0.2$ seems comparable and larger values for $\alpha$ degrade performance. Overall, we don't find goal-baseline regularization leads to better performance on these tasks.

---

**Algorithm 1** SAC with CLIP reward model.

---

**Require:** Task description $l$, encoders $\text{CLIP}_L$ and $\text{CLIP}_I$, batchsize $B$
    Initialize SAC algorithm
    $x_l \leftarrow \text{CLIP}_L(l)$                                                      ▷ Precompute task embedding
    $\mathcal{B} \leftarrow [],\ \ \mathcal{D} \leftarrow []$                                                        ▷ Initialize buffers
    **repeat**
        Sample transition $(s_t, a_t, s_{t+1})$ using current policy
        Append $(s_t, a_t, s_{t+1})$ to unlabelled buffer $\mathcal{B}$
        **if** $|\mathcal{B}| \geq |B|$ **then**
            **for** $(s_t, a_t, s_{t+1})$ in $\mathcal{B}$ **do**               ▷ In practice this loop is batched
                $x_s \leftarrow \text{CLIP}_I(\psi(s))$                  ▷ Compute state embedding
                $R_{\text{CLIP}_t} \leftarrow x_l \cdot x_s / \left(\|x_l\| \cdot \|x_s\|\right)$          ▷ Compute CLIP reward
                Optionally apply goal-baseline regularization (Definition 1)
                Remove $(s_t, a_t, s_{t+1})$ from unlabelled buffer $\mathcal{B}$
                Append $(s_t, a_t, R_{\text{CLIP}_t}, s_{t+1})$ to labelled buffer $\mathcal{D}$
        Perform standard SAC gradient step using replay buffer $\mathcal{D}$
    **until** convergence

---

for dropping the pole in `CartPole` and a positive reward for reaching the goal position in `MountainCar`. We make this change because the termination leaks information about the task completion such that without removing the termination, for example, any positive reward function will lead to the agent solving the `CartPole` task. As a result of removing early termination conditions, we make the goal state in the `MountainCar` an absorbing state of the Markov process. This is to ensure that the estimated returns are not affected by anything a policy might do after reaching the goal state. Otherwise, this could, in particular, change the optimal policy or make evaluations much noisier.

**RL Algorithms.** We use DQN (Mnih et al., 2015) for `CartPole`, our only environment with a discrete action space, and SAC (Haarnoja et al., 2018), which is designed for continuous environments, for `MountainCar`. For both algorithms, we use a standard implementation provided by `stable-baselines3` (Raffin et al., 2021).

**DQN Hyperparameters.** We train for 3 million steps with a fixed episode length of 200 steps, where we start the training after collecting 75000 steps. Every 200 steps, we perform 200 DQN updates with a learning rate of $2.3e - 3$. We save a model checkpoint every 64000 steps. The Q-networks are represented by a 2 layer MLP of width 256.

**SAC Hyperparameters.** We train for 3 million steps using SAC parameters $\tau = 0.01$, $\gamma = 0.9999$, learning rate $10^{-4}$ and entropy coefficient 0.1. The policy is represented by a 2 layer MLP of width 64. All other parameters have the default value provided by `stable-baselines3`.

We chose these hyperparameters in preliminary experiments with minimal tuning.

## C.2   HUMANOID ENVIRONMENT

For all humanoid experiments, we use SAC with the same set of hyperparameters tuned on preliminary experiments with the kneeling task. We train for 10 million steps with an episode length of 100 steps. Learning starts after 50000 initial steps and we do 100 SAC updates every 100 environment steps. We use SAC parameters $\tau = 0.005$, $\gamma = 0.95$, and learning rate $6 \cdot 10^{-4}$. We save a model checkpoint every 128000 steps. For our final evaluation, we always evaluate the checkpoint with the highest training reward. We parallelize rendering over 4 GPUs, and also use batch size $B = 3200$ for evaluating the CLIP rewards.

