# OpenReview forum: "Vision-Language Models are Zero-Shot Reward Models for Reinforcement Learning"
_ICLR.cc/2024/Conference — ICLR 2024 poster_

### Official Review · Reviewer_Q69H · 2023-10-23

**Soundness:** 3 good
**Presentation:** 3 good
**Contribution:** 3 good
**Rating:** 6
**Confidence:** 5

**Summary:**

The paper investigates using pretrained vision-language models (VLMs) as zero-shot reward models to specify tasks via natural language. Specifically, they use CLIP models to train a MuJoCo humanoid to learn complex tasks, e.g., kneeling, doing the splits, and sitting in a lotus position. Besides, they also propose a technique called goal-baseline regularization to improve performance. They also study the scaling effect of VLMs for being a reward function.

**Strengths:**

**Experiments and conclusion** The authors conduct extensive and solid experiments from the standard CartPole and MountainCar environments to the relatively complex MuJoCo humanoid environment to validate a natural idea of utilizing VLMs as a reward function to learn the agent. Besides, some in-depth studies are also well conducted, such as the scaling effect of VLMs, and various humanoid tasks specified by human language. The conclusion is interesting.

**Method**  Goal-Baseline Regularization is a novel and smart technique proposed in this work to project out irrelevant information about the observation and improve the practicability of the CLIP-based reward model.

**Writing**  The writing is clear and easy to follow; each contribution is explicitly listed and highlighted, and the section/subsection titles make navigation effortless.

**Weaknesses:**

**Real world impact**  Although the work shows the potential of VLM-RMs, all conducted experiments have been based on relatively simple and synthetic tasks, rather than from the real world. To further increase the potential impact of this work, it would be beneficial to consider some more tasks with real-world applications.

**Questions:**

When using the VLMs as the reward function for a task, one may need to consider if one should use it as a dense reward signal or a terminal reward, can the authors elaborate on how they considered this question?

---

> ### Author Response · Authors · 2023-11-16
>
> Thanks for taking the time to review our paper. Let us address the two key points: real-world impact and dense vs. sparse rewards.
>
> > Real world impact: Although the work shows the potential of VLM-RMs, all conducted experiments have been based on relatively simple and synthetic tasks, rather than from the real world.
>
> While it is true that we focused on synthetic tasks in a standard control environment, we do find evidence that VLM-RMs become significantly better the more visually realistic the environments are. Based on this, we would predict our method works even better in more practical environments. Together with the strong positive scaling with model size that we found, we think this suggests our method is highly scalable. Based on these results, we believe we demonstrate a very high potential for real-world impact, certainly compared to typical ML papers.
>
> Of course it is important to validate this hypothesis in practical environments, and we are planning to address this in future work.
>
>
> > When using the VLMs as the reward function for a task, one may need to consider if one should use it as a dense reward signal or a terminal reward, can the authors elaborate on how they considered this question?
>
> This is a good question and it is indeed not always clear how to make this choice.
>
> The main reason we chose to provide dense rewards is that we found empirically (e.g., in the classic control environments) that the VLM rewards are well shaped, i.e., measure progress towards the goal. If the rewards already have this property, we might as well provide them as dense rewards to make exploration easier.
>
> Another reason to provide dense rewards is that you might want to specify constraints that have to hold during the entire trajectory. For example we could specify that a robot should pick up an object and keep distance from a wall. Then we want to make sure to evaluate every state according to if it reaches the goal but also how close you are to the wall. And we certainly want to provide the latter as a dense reward.
>
> We hope, this clarifies any open questions about our work!

---

> > ### Author Response · Authors · 2023-11-21
> >
> > We hope our response was able to provide some additional context on your questions. We would appreciate it if you could acknowledge that you read our response and share any additional questions that come up before the discussion period ends.
> >
> > In the spirit of the discussion period, we’d be happy to dive deeper into any point of our response you disagree with. For example, we’d be interested to hear why you think the potential real-world impact of our work is limited given the strong scaling with model size and the strong benefit of realistic visuals we found?

---

### Official Review · Reviewer_13eD · 2023-10-29

**Soundness:** 2 fair
**Presentation:** 3 good
**Contribution:** 2 fair
**Rating:** 3
**Confidence:** 4

**Summary:**

The paper proposes to use pre-trained CLIP embeddings as 0-shot language-conditioned rewards for RL. It further proposes a "baseline regularization" technique that aims to normalize rewards by removing features from the reward function that are irrelevant to the task. The paper demonstrates learning of humanoid posing tasks for which ground truth rewards are hard to specify and discusses several approaches for evaluating the reward models in such scenarios without groundtruth rewards.

**Strengths:**

The problem of learning reward functions for RL is important, since for many applications, e.g. real world robot learning, it is hard to specify ground truth reward functions without serious instrumentation of the environment. The paper shows impressive results on several humanoid posing task that require a level of granularity unseen in previous 0-shot CLIP reward papers.

The proposed "baseline regularization" technique for learned rewards is interesting and novel to my knowledge. I also found the discussion of using EPIC distance to offline-score the reward model interesting.

Overall, the paper is well-written and easy to follow. I appreciate that the authors provide qualitative videos of the learned behaviors.

**Weaknesses:**

The proposed approach of using a CLIP model 0-shot to specify language-conditioned rewards is not novel, and the relevant references are cited in the related work section (Cui et al., Mahmoudieh et al.). The main technical novelty is in proposing the baseline regularization. However, the main experiment on learning humanoid posing tasks does not use this regularization at all. As such, it appears that the used approach in these experiments is identical to those proposed in prior work (Mahmoudieh et al), just applied on a different task.

In the same vein, the paper is lacking any comparison to prior work. This may be because the method is identical to prior work, just applied on a different domain? I do think it is valuable to show that CLIP-based rewards can work 0-shot in more challenging domains, but without technical novelty this contribution seems insufficient to warrant acceptance.

Figure 4 shows that the proposed technical novelty of the baseline normalization does improve EPIC distance between the learned and a human-provided reward function. However, the paper lacks evidence that this improvement in EPIC distance indeed translates in an improvement in policy performance once trained on the reward function (the performed evaluation is a single step function from 0% to 100% success, so it's not informative enough to show good correlation, an environment with a more finegrained reward function may be required).

A minor weakness is, that the experiments were performed in a clean, simulated environment, without distractors or other moving objects. Thus it is unclear whether the reward functions would be robust to such distractors (though admittedly prior works also did not evaluate realistic scenes).

Finally, albeit impressive, the experiments are only performed on one non-toy environment (the humanoid). So even if the authors demonstrated that the proposed baseline regularization helps to improve policy performance, it would be good to add evaluations on at least a second non-toy environment, e.g. a robotic manipulation task, to prove that the effects are consistent.

**Questions:**

- how does the evaluated approach for Table 1 differ from prior works like Mahmoudieh et al that also use 0-shot CLIP embeddings for language-conditioned reward computation?

- in Mahmoudieh et al. the non-finetuned model does not work at all as a reward function -- is the fact that the model in the submission works well purely based on the chosen task domain?


## Summary of Review
Overall, the paper shows some interesting learned behaviors with a simple method, but the novelty over prior work is not well demonstrated in the experimental results. No comparisons to baselines are performed and the introduced algorithmic changes are not used in the experiments. As such, I do not recommend acceptance of the paper in it's current form. I do think the paper has potential and I encourage the authors to
- add experimental results that demonstrate that the proposed baseline regularization leads to improved policy performance
- add experiments on at least one additional non-toy domain, e.g. a robotic manipulation task
- demonstrate experimentally the correlation between the offline EPIC reward distance metric and policy performance once trained on the reward function

# Post-Rebuttal Comments

Thank you for answering my review!
I want to emphasize that I do not devalue empirical works in any way -- there are lots of examples for very high impact empirical works that shed light on shortcomings of the existing literature and provide practical implementation advice to "make things work".

The critical elements for such papers are that they provide technical "tricks" that are specific to the core algorithm yet demonstrate that they help across a wide range of applications of this algorithm (ie are not overfit to one task / environment etc.).

Based the on the rebuttal the paper claims three main “tricks” to make things work:
   - (1) model size —> this one makes sense and is validated with experiments
   - (2) reward formulation —> this one is not validated experimentally
   - (3) choice of RL optimizer algorithm —> this one seems orthogonal to the core question of how to formulate reward functions and is also not experimentally validated

Further, I reiterate my concern that all experiments are conducted in a single non-toy environment — this is in my opinion insufficient to claim that the paper introduces “a general way to make CLIP-rewards work”.

If the main scope of the paper should be “tricks to make CLIP-rewards work”, the writing would need to change substantially, the experimental evaluations mentioned above would need to be added and more non-toy enviornments would need to be evaluated.

Finally, it remains confusing that the main technical contribution is not used in the main experimental evaluation and the rebuttal did not provide a convincing reason for this.

Thus, in summary I do not see my concerns addressed by the rebuttal and maintain my score.

**Details Of Ethics Concerns:**

--

---

> ### Author Response · Authors · 2023-11-16
>
> Thanks for this detailed review! Your comments will certainly help to improve our paper!
>
> Before answering specific questions, let us make a general point on novelty. While our paper does not propose an entirely new method, it is the first paper to make this method work for learning complex tasks. This positive result which stands in contrast to previous weaker negative results is an important contribution and might make a difference in whether people actually use this method or not. The goal of empirical research in ML should be to produce evidence on which methods work and not only to develop new methods.
>
> > How does the evaluated approach for Table 1 differ from prior works like Mahmoudieh et al that also use 0-shot CLIP embeddings for language-conditioned reward computation?
>
> The primary method proposed by Mahmoudieh et al. uses task-specific finetuning. This makes it much more expensive and less general than the zero-shot method we focus on. They do briefly consider a method they call “Vanilla Dot-product Visuolinguistic Base Reward Model” as a baseline which is similar to ours.
>
> Importantly Mahmoudieh et al. conclude this baseline does not work well, motivating the design of their more task-specific method. For example, in Section 2.1. they say: “ One significant limitation of CLIP is that it cannot distinguish spatial relationship of objects in images. This limits our base reward model from being useful for tasks that have spatial goals. Our full zero-shot reward model remedies this issue by leveraging CLIP in a very different way.”
>
> Unfortunately, Mahmoudieh et al. do not provide code to reproduce the experiments. But from the paper, three key differences stick out:
> - **Reward formulation**: Their base reward model uses an (unnormalized) dot product as a reward, whereas we use a cosine similarity (i.e., a normalized dot product). Cosine similarity is more aligned with the CLIP training objective, so it should work better.
> - **Model size**: They use the RN50 CLIP model, which is the smallest model out of the ones we use. One of our key findings is that model size is very important for CLIP models being useful zero-shot reward models.
> - **Training**: They train for 200k steps, whereas we train for 10M steps. We also use SAC over PPO, and the former is typically more sample efficient.
>
> > In Mahmoudieh et al. the non-finetuned model does not work at all as a reward function -- is the fact that the model in the submission works well purely based on the chosen task domain?
>
> We suspect that all of the above mentioned differences are important for our method to work well. Specifically, we show in Figure 4 that model size is critically important, and we observed that for less than 1M training steps we see barely any training progress. While we did not test the (unnormalized) dot-product formulation systematically, early experiments suggested that it might be more noisy. Certainly, it is conceptually less principled because CLIP is trained with a cosine similarity objective.
>
> We would have liked to evaluate our method in the environments studied by Mahmoudieh et al. Unfortunately, they use non-standard tasks and do not provide code, so it would be difficult to reproduce their setup without significant effort. We agree that it is an open question how well VLM-RMs will work in robotics domains where many tasks are of a more “spatial” nature, and we are actively interested in studying this in future work.
>
>
> > In the same vein, the paper is lacking any comparison to prior work. This may be because the method is identical to prior work, just applied on a different domain? I do think it is valuable to show that CLIP-based rewards can work 0-shot in more challenging domains, but without technical novelty this contribution seems insufficient to warrant acceptance.
>
> Our focus is on tasks that do not have a ground truth reward function, and as such we rely on human evaluation. The reason for this is to demonstrate we can learn tasks with VLMs that it would be difficult to impossible to specify reward functions for. As far as we can tell, our method is the first to reliably achieve this goal.
>
> Note that we do show that our method contribution (goal-baseline regularization) helps significantly with reward shaping in classic control tasks (Fig 2) and consistently improves the EPIC distance between the reward model and human labels in the humanoid (Fig 4).
>
> We see our main contribution with this work as demonstrating that a method that was previously thought to perform poorly, actually performs well when implemented carefully and scaled up. We provide a lot of new empirical insights primarily about scalability and evaluating tasks without ground truth rewards. We think such work is of high value to the ML community and should be particularly interesting to the empirically minded ICLR community.
>
> We hope that this clarified some of the key contributions of our paper. Please follow-up if any further questions come up.

---

> > ### Author Response · Authors · 2023-11-21
> >
> > We hope our response was able to provide some additional information about your questions. We would appreciate it if you could acknowledge that you read our response and share any additional questions that come up before the discussion period ends.
> >
> > Given our comment clarifying the contributions and novelty of the paper, we would be interested if your reasons for rejecting our paper are primarily that you think the technical work is low quality, or because you think the novelty of the method is too limited?

---

### Official Review · Reviewer_rUuN · 2023-10-30

**Soundness:** 2 fair
**Presentation:** 2 fair
**Contribution:** 2 fair
**Rating:** 3
**Confidence:** 4

**Summary:**

Using VLM to generate intrinsic rewards is a new venue since manually specifying reward functions for real-world tasks is often infeasible. In this paper, the authors propose VLM-RM, a method for using pre-trained VLMs as a reward model for vision-based RL tasks. In more detail, they introduce a baseline as regularization for the final reward. The baseline aims to remove the irrelevant part in the CLIP embeddings. This irrelevant info means the natural language description of the environment setting in its default state, irrespective of the goal. As for the experiment part, the authors validate their method in the standard CartPole and MountainCar RL benchmarks.

**Strengths:**

1. Designing a more RL-friendly reward is an interesting research direction. It can help the generalization of policy learning.
2. Removing the unimportant part from the CLIP embeddings seems to be a reasonable direction for improving the performance of VLM rewards.
3. The paper is easy to follow.

**Weaknesses:**

1. The contribution is limited and the motivation is not clear. Intuitively, I can also claim that the background information of the language description is not useless since different instructions may have different meanings in different environments. I encourage authors to provide more theoretical analysis to support the effectiveness of the proposed baseline if any. In addition, simply proposing one regularization is not novelty enough unless the authors can prove it can boost performance in a wide range of tasks.

2.  The experiments are not sufficient. Taking Figure 3 as an example, more seeds and more tasks should be included for baselines.

**Questions:**

None.

---

> ### Author Response · Authors · 2023-11-16
>
> Thanks for the review. We will aim to provide some context on the key concerns in the review. It would be valuable, if you could provide more specific criticisms or questions for us to respond to.
>
> > The contribution is limited and the motivation is not clear. Intuitively, I can also claim that the background information of the language description is not useless since different instructions may have different meanings in different environments.
>
> **Motivation**: Training RL agents in environments where reward functions are difficult to obtain is an important and practical research problem, with many applications from language models to real-world robotics. Reward functions may not be available because we are operating in the real world as opposed to a simulation, and many tasks do not have a clear ground truth reward function at all. Learning reward models from human data is common practice to address this problem in many domains, but it requires a large amount of expensive human data. We believe that any way to reduce the amount of data necessary is highly relevant and important to solve practical problems. It is true that language instructions can sometimes be ambiguous, but this is no reason to discard this extremely valuable source of reward information. We believe language specification can heavily reduce the amount of explicit human preference data necessary, providing ample motivation for our work.
>
> **Contribution**: To the best of our knowledge, our paper is the first to show that VLM-RMs can work reliably over a wider range of tasks in a complex environment, contradicting previous weak evidence of the contrary. We provide multiple technical contributions, including a “method” contribution (goal-baseline regularization) and many novel insights into how well VLMs work as reward models. We show how their rewards are shaped, how they compare to human labels, and, most importantly, how their performance scales with model size (a key insight missing from previous work, which at the time seemed to show the vanilla VLM-RM approach not to work). **Could you please clarify why you find the contribution limited?**
>
>
> > The experiments are not sufficient. Taking Figure 3 as an example, more seeds and more tasks should be included for baselines.
>
> We focus on learning tasks that do not have a ground truth reward function, so we need human evaluation. While we agree that more seeds and tasks would be desirable, evaluation tends to be quite expensive in our setting.
>
> Specifically, note that Figure 3 is an ablation of some features of our setup but still requires human evaluation. Running this ablation for many tasks and seeds would be very expensive and likely provide limited new insights.
>
> We think we did quite extensive experiments in simple and complex environments. For example, we thoroughly studied effects of different model sizes and goal-baseline regularization and performed different ablations. **Could you please clarify specifically which parts of our experiments you find lacking?**
>
> We hope the response addresses some of your concerns, and we would appreciate any clarification of your specific concerns, to allow us to respond to them.

---

> > ### Author Response · Authors · 2023-11-21
> > **Please clarify your concerns about our paper**
> >
> > Dear reviewer rUuN, in the interest of a productive discussion period, could you please be more specific in your criticisms of our paper? Unfortunately, from your current review it does not become clear:
> >
> > - Which specific weaknesses of our paper make you think the paper should be rejected?
> > - Why do you find the contribution limited?
> > - Why do you find the motivation not clear?
> > - Which additional experiments would you like to see (beyond “more tasks” and “more seeds”)?
> >
> > If you can clarify these points, we can provide a much clearer answer to elaborate on our position about your concerns.

---

> > > ### Comment · Reviewer_rUuN · 2023-11-21
> > >
> > > Apologizing for late replying.
> > >
> > > As for contribution, I admit the proposed method is somewhat interesting, however, I still do not believe it reaches the bar of ICLR. This paper wants to remove the irrelevant parts from the environment if I understand correctly. However, how can you guarantee that the environmental part cannot be useful to distinguish the state and design reward in other cases? For example, moving forward in the water and moving forward on the ground definitely mean two different behaviors. Therefore, as I mentioned, I encourage authors to provide more theoretical analysis to support the effectiveness of the proposed method if any. If there is only one method contribution without solid motivation, I am not convinced.
> > >
> > > If I understand wrongly, I can reconsider the score.
> > >
> > > As for experiments, reporting a success rate is meaningless if no confidence interval is reported. Now I understand that human evaluation would be very expensive, therefore I would not ask for more seeds.
> > >
> > > However, I found Reviewer 13eD raised some points, such as adding experimental results that demonstrate that the proposed baseline regularization leads to improved policy performance and adding experiments on at least one additional non-toy domain, e.g. a robotic manipulation task. I agree with the reviewer for these parts and I didn't find any modification regarding these parts.
> > >
> > > Lastly, as for the contribution standard of ICLR, I would take another decent paper, Eureka: Human-Level Reward Design via Coding Large Language Models Reward Design via Coding Large Language Models, as an example. I know these two works share different motivations, but they all want to design the reward automatically. As you can check in the open review, that work was interesting in both depth and breadth but still got borderline initially.

---

> > > > ### Author Response · Authors · 2023-11-21
> > > >
> > > > Thank you for your response and for further clarifying your opinion on our paper. We believe we have satisfying answers to each concern you expressed.
> > > >
> > > > > As for contribution, I admit the proposed method is somewhat interesting, however, I still do not believe it reaches the bar of ICLR. This paper wants to remove the irrelevant parts from the environment if I understand correctly.
> > > >
> > > > We want to further clarify that “goal-baseline regularization”, while the main methodological novelty, is only part of our paper’s contribution.
> > > >
> > > > Fundamentally, our paper is an empirical study of Vision-Language Models as Reward Models _in a zero-shot manner_. Our results include novel and impactful insights _even for the most basic VLM-RM approach_ without goal-baseline regularization. We show in our work that, without expensive fine-tuning or using other complex techniques, Vision-Language Model rewards can be **well-shaped, aligned with human labels, and scale predictably** in quality with size.
> > > >
> > > > Previous work, such as Mahmoudieh et al., has only tried similar methods with small CLIP methods, and found them to not work robustly. In contrast, we show that to obtain useful rewards, the most important variable is model size. We find it extremely important to report these findings to demonstrate that VLM reward models can work well at larger scales previously not investigated.
> > > >
> > > > Most importantly, **we are the first to show how to learn complex tasks with VLM-RMs that do not have a ground truth reward function**!
> > > >
> > > > Goal-baseline regularization is a novel, simple method for obtaining higher performance from VLM-RMs in a zero-shot manner without adding too much complexity. While it’s an important contribution of the paper, we think it has to be considered in the context of the points above.
> > > >
> > > > We hope that this clarifies the scope of the paper’s contributions, potentially leading the reviewer to reconsider their score.
> > > >
> > > > > As for experiments, reporting a success rate is meaningless if no confidence interval is reported.
> > > >
> > > > We evaluate 100 trajectory from each policy. Figures 5 and 6 in Appendix C show histograms for each evaluation which show the variation of the human labels. We will make sure to point to this figure from the main paper and add an aggregated standard error to the tables. We hope that this addresses the concern about the results’ reliability.
> > > >
> > > > > Lastly, as for the contribution standard of ICLR, I would take another decent paper, Eureka: Human-Level Reward Design via Coding Large Language Models Reward Design via Coding Large Language Models, as an example. I know these two works share different motivations, but they all want to design the reward automatically. As you can check in the open review, that work was interesting in both depth and breadth but still got borderline initially.
> > > >
> > > > We would push back on this last point on two fronts:
> > > > 1. We do not understand the argument related to the reviews obtained by the Eureka paper. Both papers should be judged on their own merits. Specifically, a paper the reviewer likes receiving low scores should not be a reason for giving low scores to another paper.
> > > > 2. The Eureka paper involves using language models to write code for reward functions in simulated environments, where the internal state of the simulation can be used to craft a well shaped reward function. In contrast, we use VLMs to obtain reward signals _directly from image observations_, rendering our method more generally suitable for real world tasks. We would argue that the motivation of both works are very similar (obtain rewards from language prompts), but the methods and papers are very different. So even if there was an argument to be made by comparing different papers' scores, these specific two papers are sufficiently different that we do not think this comparison is useful.

---

> > > > > ### Comment · Reviewer_rUuN · 2023-11-22
> > > > >
> > > > > Thank you for the quick response which addresses some of my concerns, I have two questions left.
> > > > > 1. For the claim: we are the first to show how to learn complex tasks with VLM-RMs that do not have a ground truth reward function, what do you think is the difference between your work and Minedojo, VLM model that generates a reward for the Minecraft. Why cannot they be considered as the first work?
> > > > >
> > > > > 2. You still haven't replied to me about why you ignored the comment that the reviewer 13eD encouraged you to add more experiments. I understand the reviewer cannot ask for everything, but a reason is required.

---

> > > > > > ### Author Response · Authors · 2023-11-22
> > > > > >
> > > > > > > For the claim: we are the first to show how to learn complex tasks with VLM-RMs that do not have a ground truth reward function, what do you think is the difference between your work and Minedojo, VLM model that generates a reward for the Minecraft. Why cannot they be considered as the first work?
> > > > > >
> > > > > > The Minedojo paper [1] proposes MineCLIP, which indeed uses a CLIP model as a reward model. However, they _train_ this model on a lot of Minecraft-specific data (specifically, "640K pairs of 16-second video snippets and time-aligned English transcripts", cf. page 8 in [1]). Our previous statement was referring to being the first to successfully use VLMs as **zero-shot** reward models, i.e., without any training.
> > > > > >
> > > > > > As stated in our previous comment, our paper shows that **without expensive fine-tuning** or using other complex techniques, Vision-Language Model rewards can be well shaped, aligned with human labels, and most importantly, scale predictably in quality with size.
> > > > > >
> > > > > > Given that MineCLIP uses expensive, task-specific fine-tuning, it does not provide information on any of these points.
> > > > > >
> > > > > > We agree the reference [1] is missing in our related work section, and we will add it when revising the paper.
> > > > > >
> > > > > > [1] Fan, Linxi, et al. "Minedojo: Building open-ended embodied agents with internet-scale knowledge." Advances in Neural Information Processing Systems 35 (2022): 18343-18362.
> > > > > >
> > > > > >
> > > > > > > You still haven't replied to me about why you ignored the comment that the reviewer 13eD encouraged you to add more experiments. I understand the reviewer cannot ask for everything, but a reason is required.
> > > > > >
> > > > > > We apologize for seemingly ignoring this concern. We were under the impression that the other points were more central to your concern. We’re happy to comment on reviewer 13eD’s suggested additional experiment.
> > > > > >
> > > > > > > adding experimental results that demonstrate that the proposed baseline regularization leads to improved policy performance
> > > > > >
> > > > > > For the simple control benchmarks of Cartpole and MountainCar, we tested goal-baseline regularization extensively.
> > > > > >
> > > > > > We can solve the Cartpole environment without any regularization, and regularization does neither help nor hurt performance. However we cannot solve the MountainCar environment without using goal-baseline regularization.
> > > > > >
> > > > > > We did not originally present training results in detail, because we expected the reward shape plots in Figure 2 (b) and (c) sufficiently show the advantage of goal-baseline regularization in the Mountain car. But given the present discussion, we will add results from training too.
> > > > > >
> > > > > > For the humanoid tasks, we demonstrate that goal-baseline regularization consistently improves the EPIC distance between the reward model and human labels in the humanoid (see Figure 4a). Unfortunately, the human evaluation makes it infeasible to evaluate policies trained for many different values of alpha, and certainly we would not be able to provide these results within the rebuttal period.
> > > > > >
> > > > > > > adding experiments on at least one additional non-toy domain, e.g. a robotic manipulation task
> > > > > >
> > > > > > We agree that moving to more practical tasks is an important next step, and we are working on follow-up work to fully address this.
> > > > > >
> > > > > > However, we disagree that simulated robotic manipulation tasks are automatically “non-toy domains” compared to the humanoid environment. Many standard robotics environments do not allow for tasks that require complex reward functions, which was the focus of our work, while not being harder from a control perspective than the humanoid (which requires controlling 16 joints).
> > > > > >
> > > > > > Setting up more interesting robotics environments is an important direction, but requires careful thought and effort beyond the scope of this paper. Specifically, providing meaningful results in additional environments would not have been feasible to run within the rebuttal period.
> > > > > >
> > > > > > ------
> > > > > >
> > > > > > Thanks for engaging in the discussion period. Please let us now if any further questions come up.  If not, we hope we were able to address your concerns, and that you might consider raising your score.

---

### Official Review · Reviewer_CqGG · 2023-10-30

**Soundness:** 3 good
**Presentation:** 3 good
**Contribution:** 3 good
**Rating:** 8
**Confidence:** 4

**Summary:**

This paper investigates the use of pre-trained visual-language models for specifying reward models for reinforcement learning agents via natural language. In its simplest form, a single sentence text prompt is passed to a CLIP model along with the pixel observation of the agent to produce a score that translates to the reward. A more sophisticated method is also proposed, labelled goal-baseline regularisation, with the intention of removing irrelevant information about the environment from the observation embedding. Specifically, it involves projecting the observation embedding on to the the line spanned by the task embedding and a baseline embedding, the latter of which originates from a generic description of the environment. The CLIP alignment is then calculated using an interpolation of the original observation embedding and the projection embedding. The approach is validated in the context of two simpler task, CartPole and MountainCar, as well as more complex ones in the Humanoid environment. For the first two, it is shown that the CLIP-induced reward aligns well with the ground-truth reward (which is not provided to the agent) and that agents trained with the former can learn to perform well in the initial task. In the Humanoid environment, agents are trained on complex tasks with no available ground-truth reward, and are shown to perform well in most tasks as judged by a human evaluation. It is shown that modifying the backgrounds and textures of the visual observations to make them more realistic can have a large effect on the success of agents. Furthermore, scaling experiments are run that demonstrate that using more powerful VLMs leads to better agent performance.

**Strengths:**

- The paper tackles the important and challenging problem of reward specification in reinforcement learning and effectively demonstrates a promising approach using VLMs.
- The method is clearly explained and theoretically simple to implement.
- There is valuable insight derived from the ‘tricks’ that get the method to work, namely the goal-baseline regularisation and the re-rendering of the observations to make them more realistic.
- Interesting idea to evaluate performance on tasks with no ground truth reward via EPIC distance with human-evaluated reward.
- The scaling experiments that show that performance improves with the size of the VLM are very encouraging for the scalability of the method.

**Weaknesses:**

- Risk of bias in human experiments. While this is duly acknowledged in the paper, the fact that the human analysis was conducted by one of the authors of the paper is a potentially strong source of bias in the human evaluation.
- It is stated that ‘minimal prompt engineering’ was required to find the right single text prompt for the reward function but the process for discovering the right prompt and the robustness of the method with respect to noise in the prompt / semantic variations is not discussed.
- There is some evidence in the experiments of reward misspecification with the VLM, a theoretical limitation acknowledged by the authors in the conclusion. It is noted that when the regularisation strength is high for realistic MountainCar that the CLIP reward function reflects the shape of the mountain. The fact that going up the small hill to the left gives high CLIP reward is useful because it encourages a policy where the mountain car oscillates in the valley until it has enough momentum to reach the top right (where the ground truth reward lies), but this is somewhat of a fortunate coincidence as there is actually a mismatch here with the ground-truth reward, which only rewards reaching the top right.

**Questions:**

- Could the authors elaborate on the ‘minimal prompt engineering’ required to find the single sentence prompts for each task? Also, are there any results demonstrating the robustness of the method to syntactic variations in the prompt?
- Suggested citation if authors believe it is relevant (using VLMs to improve exploration in RL): https://openreview.net/forum?id=-NOQJw5z_KY

---

> ### Author Response · Authors · 2023-11-16
>
> Thanks for the valuable review, we will address the remaining open questions one by one.
>
> > Risk of bias in human experiments While this is duly acknowledged in the paper, the fact that the human analysis was conducted by one of the authors of the paper is a potentially strong source of bias in the human evaluation.
>
> This is true and a valid criticism. To reduce bias in the evaluation, we will publish the full raw videos we evaluated to produce the numerical results together with the paper (rather than only the samples currently provided on the website).
>
>
> > There is some evidence in the experiments of reward misspecification with the VLM, a theoretical limitation acknowledged by the authors in the conclusion. It is noted that when the regularisation strength is high for realistic MountainCar that the CLIP reward function reflects the shape of the mountain. The fact that going up the small hill to the left gives high CLIP reward is useful because it encourages a policy where the mountain car oscillates in the valley until it has enough momentum to reach the top right (where the ground truth reward lies), but this is somewhat of a fortunate coincidence as there is actually a mismatch here with the ground-truth reward, which only rewards reaching the top right.
>
> We note that in the MountainCar, our goal prompt is “a car at the peak of the mountain, next to the yellow flag”. Given this prompt, it would make sense for reward signals to encode both being on top of a mountain and being close to the flag. While the reward landscape follows the shape of the mountains, the peak on the right mountain is bigger than the left mountain (even adjusting for their relative height). So, the reward does seem to capture both the mountain height and the flag position in this case.
>
> That being said, we don’t yet have a good sense of how problematic misspecification with VLM-RMs is in practice. But it is certainly an important question and we plan to investigate this in more detail in follow-up work.
>
>
> > Could the authors elaborate on the ‘minimal prompt engineering’ required to find the single sentence prompts for each task?
>
> In early experiments we tried a few prompt variations for the “kneeling” tasks and did not find much variation. For the other tasks, we only tried the single prompt that is provided in the paper. Only for the “standing on one leg” task, we tried a few prompts to see if we can get better performance, but saw no improvement. We will clarify this "minimal prompt engineering" in the paper.
>
> Overall, we think it's fair to say our results required essentially zero prompt engineering. The experiments we ran with different prompts further suggest that results are quite insensitive to prompt variations.
>
>
> > Also, are there any results demonstrating the robustness of the method to syntactic variations in the prompt?
>
> We tried averaging the reward over synthetic prompt variations in early experiments for the “kneeling” task but did not find noticeable improvements in robustness, so we went for the simplest version of the algorithm for the final training runs. But it is quite possible that we used prompts with too simple of a syntactic structure and alternative versions of this method would yield more improvements. This would certainly be interesting to explore more in future work!
>
>
>
> > Suggested citation if authors believe it is relevant (using VLMs to improve exploration in RL): https://openreview.net/forum?id=-NOQJw5z_KY
>
> Thanks for the suggestions. We will add it as useful evidence on VLM representations providing useful information for RL algorithms.

---

> > ### Author Response · Authors · 2023-11-21
> > **Raw data from human evaluation**
> >
> > We have now uploaded the raw data from our human evaluations here: https://drive.google.com/file/d/1FhjS5UGg4g6GzPUHHZ-FdgGyCTd9YOq9/view We will add this link to the final version of the paper.
> >
> > For each policy we evaluate, the data contains 100 sample trajectories and the corresponding human success labels as described in the paper. This will allow interested readers to spot check the labels and run other analyses on this data.
> >
> > We're happy to answer any questions that might have come up in the meantime!

---

### Official Review · Reviewer_kRKZ · 2023-11-01

**Soundness:** 4 excellent
**Presentation:** 4 excellent
**Contribution:** 4 excellent
**Rating:** 8
**Confidence:** 4

**Summary:**

This paper proposes the VLM-RM method, which uses a pre-trained CLIP model as a reward for training vision-based agents. VLM-RM models the reward as the alignment between a language instruction specifying the task and the current observation image while regularizing relative to a baseline prompt to remove irrelevant information from the reward. VLM-RM is empirically validated in classic control tasks and reaching various poses in MuJoCo.

**Strengths:**

- The proposed VLM-RM method is simple and works well. VLM-RM provides a way to use CLIP models easily as a zero-shot reward signal from text without having to finetune the CLIP model. Furthermore, the prompts used for the reward model in VLM-RM are simple and intuitive, highlighting the ease of applying VLM-RM.
- The experiments show the applicability of VLM-RM. In CartPole and MountainCar, the CLIP reward predictions align with the true success state, and we see the goal-baseline regularization helping for MountainCar. VLM-RM is also able to learn a variety of behaviors in the high-dimensional humanoid task. While the success of the humanoid experiments was evaluated by just one of the authors, the videos on the website convincingly show the correct behaviors.
- The work clearly explores the limitation of VLM-RM that the environments should be visually realistic for CLIP to provide a meaningful learning signal. In MountainCar, realism produces better alignment between CLIP and the true success state (Fig. 2c). In Fig. 3, we see the impact of modified textures and the camera placement on performance. I believe the limitation around the realism of the observations is more a limitation of the environments rather than VLM-RM.
- The work shows evidence that VLM-RM scales to better performance with larger, more capable CLIP models. Fig. 4, shows the humanoid kneeling task is only possible with the largest CLIP model.

**Weaknesses:**

- The paper states that the "CLIP rewards are only meaningful and well-shaped for environments that are photorealistic enough for the CLIP visual encoder to interpret correctly," yet the paper focuses on control environments without realistic visuals. Why not instead focus on more visually realistic simulation benchmarks and not have to modify the environment to make the rendering more realistic to fit the algorithm?
- It's unclear if the goal-baseline regularization is necessary. The primary experiments in Table 1 don't use any goal-baseline regularization. The analysis in Fig. 4a shows the maximum or minimum amount of goal regularization to be best. Additionally, to what degree can the same effects of the goal-baseline regularization be incorporated in the goal prompt? A more detailed goal prompt can just specify to ignore the irrelevant information.
- While the paper shows the strong final results of VLM-RM, the RL training stability of VLM-RM is unclear. Is it easy for RL methods to learn from the VLM-RM reward? How does this compare to a ground truth reward? See my further comments under the questions section.

**Questions:**

- What exact values are selected for $\alpha$ in Fig. 4a? I recommend putting an indicator in this plot to show which values are used.
- What is the optional context $c$ in Eq. 1 used for? I don't see it referred to later in the paper.
- Can the paper include the RL learning curves for all the experiments showing the number of samples versus either the predicted CLIP return or a ground truth metric, like true reward in the classic control tasks or EPIC in the humanoid tasks? Given there is a ground truth reward in the classic control tasks, can the authors also include that as a reference for the VLM-RM learning curves? It is important to see the training stability and efficiency under the CLIP model reward.

---

> ### Author Response · Authors · 2023-11-16
>
> Thanks for the useful and positive feedback. The review raises a couple of great questions that we’ll address one by one.
>
> > Why not instead focus on more visually realistic simulation benchmarks and not have to modify the environment to make the rendering more realistic to fit the algorithm?
>
> We found the MuJoCo humanoid environment to be a good starting point because it is a standard control benchmark and relatively inexpensive to set up, run and render. This environment provides a fast feedback loop to test different aspects of the algorithm (like scale and baseline prompts) but it is still complex enough to allow for interesting tasks. We discovered only through our experiments that realism of the visual representation is quite important. Based on our results, we would predict that VLM-RMs work even better in visually more realistic environments and we’re planning to test this in follow-up work.
>
> > It's unclear if the goal-baseline regularization is necessary. The primary experiments in Table 1 don't use any goal-baseline regularization. The analysis in Fig. 4a shows the maximum or minimum amount of goal regularization to be best.
>
> Recall that lower EPIC distance indicates a better correlation with the human labels, i.e., a better reward model. Fig. 4a suggests that typically a regularization strength between 0 and 1 is better than the extremes 0 or 1. Only for RN50 (the smallest CLIP model we tried), the maximum amount of regularization seems to be best. Importantly, in none of the cases the minimum amount of regularization seems to be best.
>
> However, the effect of the regularization is weaker for larger models, and the difference in our results is not always significant. We interpret this to mean that goal regularization can help a lot with reward shaping when using smaller models, but might be unnecessary when using large enough models.
>
>
> > Additionally, to what degree can the same effects of the goal-baseline regularization be incorporated in the goal prompt? A more detailed goal prompt can just specify to ignore the irrelevant information.
>
> This is a great question! In principle, we can encode all necessary information in the prompt. In practice, this is difficult with current models. For example, it’s well documented that current CLIP models can struggle with negations which would be necessary to describe the contrast between the goal and the baseline prompt. Hence, it is better to use the embeddings of a goal and baseline prompt and encode the negation as a subtraction as in our goal-baseline regularization. But we do expect future VLMs to get better at handling negations and other complex prompts.
>
> > What exact values are selected for in Fig. 4a?
>
> Good point, we’ll add indicators to the figure. The values for alpha are: 0.0, 0.1, 0.2, 0.3, 0.4, 0.5, 0.6, 0.7, 0.8, 0.9, 0.99.
>
> > What is the optional context in Eq. 1 used for?
>
> In general, we define the context as a “catch-all” term containing information provided to the VLM that is not the goal prompt. In our case this only happens via the baseline regularization technique. So, you can think of the context in our implementation as containing the baseline prompt and regularization strength. We’ll clarify this in the paper. In other kinds of VLM-RMs involving vision-augmented language models (not yet tested by us), the context could include a system prompt, used to make the model to act as a reward model.
>
> > Can the paper include the RL learning curves for all the experiments showing the number of samples versus either the predicted CLIP return or a ground truth metric, like true reward in the classic control tasks or EPIC in the humanoid tasks? Given there is a ground truth reward in the classic control tasks, can the authors also include that as a reference for the VLM-RM learning curves?
>
> We can plot learning curves of the CLIP rewards, but, as you say, they don’t necessarily show actual performance. While we can evaluate the ground truth reward in the classical control tasks, this is not possible for the humanoid tasks where we specifically chose “semantic” tasks with no programmatic ground truth function.
>
> EPIC distance is only a measure of distance between human reward labels and the CLIP reward labels, but it also does not allow us to show the policies’ learning progress during training.
>
> That being said, we agree it would be good to provide some learning curves in the paper to evaluate training stability, and we will do so (we will add them to the appendix). In brief: we find some variation depending on the initialization of the policy (for some random seeds the policies learn poorly), but overall training is relatively stable and comparable to the ground truth reward for task where we have one.

---

> > ### Author Response · Authors · 2023-11-22
> > **Training curves for all experiments**
> >
> > For now, we have uploaded a PDF containing training curves for all experiments at the following link: https://drive.google.com/file/d/1W1ERoHZugbpi0Ap2yC45qjtf8PryT11x/view
> >
> > As you can see the training is reasonably stable, and in cases where we have ground truth reward functions, the CLIP training curve closely mirrors the ground truth training curve. When revising the paper, we will add these plots and make sure to point to the training curves when discussing training stability.

---

### Public Comment · ~Zihan_Ding1 · 2023-11-14
**A quick comment by reader**

This is an interesting paper drawing my attention.

Some quick clarification questions about Fig. 2 results:
* How is the rescaling achieved and is it also used in later Humanoid tasks?
* Which VLM (size) is used for Fig. 2?

Thanks

---

> ### Author Response · Authors · 2023-11-16
>
> Thanks for your interest in our paper!
>
> > How is the rescaling achieved and is it also used in later Humanoid tasks?
>
> The rescaling is only for visualization purposes in the plot, and is not used at all during training. For each value of $\alpha$, we normalize each data series to be between 0 and 1, i.e.  $r_\mathrm{norm} = (r - r_\mathrm{min}) / (r_\mathrm{max} - r_\mathrm{min})$, to make it easier to visually compare the reward shape for different values of alpha.
>
> > Which VLM (size) is used for Fig. 2?
>
> This figure uses the largest model we considered: ViT-bigG-14

---

### Meta-Review · Area_Chair_ZMfx · 2023-12-05

**Metareview:**

This paper uses a VLM based reward model as a zero shot reward model for training vision-based agents for classical control environments up to MuJoCo.

Despite some negative reviews, I recommend accepting this paper:
- The proposed method is simple and works well. In particular, a CLIP model can be used out of the box.
- The experimental results seem impressive and are likely of interest to the community.
- The paper documents evidence that shows that larger and larger reward models improve performance.

**Justification For Why Not Higher Score:**

This paper is of interest to the community but not exceptional, in that it merits a spotlight.

**Justification For Why Not Lower Score:**

Despite some negative reviews, I recommend accepting this paper:
- The proposed method is simple and works well. In particular, a CLIP model can be used out of the box.
- The experimental results seem impressive and are likely of interest to the community.
- The paper documents evidence that shows that larger and larger reward models improve performance.

---

### Decision · Program_Chairs · 2024-01-16

Accept (poster)